# Associations between systemic inflammation, nutritional status, and cardiometabolic diseases and risk factors among adults living in transitional rural communities in Ecuador

Irina Chis Ster[1], Monsermin Gualan[2,3], Luz-Marina Llangari-Arizo[2,3], Andrea Lopez[2], Natalia Romero-Sandoval[2,3], Philip J. Cooper[1,2*]

1 School of Health and Medical Sciences, City St George's University of London, London, United Kingdom, 2 Universidad Internacional del Ecuador UIDE, Quito, Ecuador, 3 Grup's de Recerca d' Amèrica i Àfrica Llatines-GRAAL, Cerdanyola del Valles, Barcelona, Spain

* pcooper@citystgeorges.ac.uk

## Abstract

### Background

Chronic low-grade systemic inflammation, as indicated by elevated high-sensitivity C-reactive protein (hs-CRP), has been implicated in the pathogenesis of cardiometabolic diseases. Less is known about the role of total immunoglobulin E (IgE), a marker of type 2 T helper cell-driven inflammation, in such outcomes. Information on the relevance of these markers in rural populations in transitional rural communities remains scarce, despite their growing burden of non-communicable diseases.

### Objectives

This study examined the associations between two inflammatory markers—CRP and total IgE—and cardiometabolic risk factors and diseases in adults from transitional rural communities in coastal Ecuador.

### Methods

We conducted a cross-sectional analysis of 931 adults from ten rural agricultural communities. Standardized questionnaires, anthropometric measurements, and fasting blood samples were used to collect data on sociodemographics, body composition, biochemical risk factors, and inflammatory markers (CRP and total IgE). Cardiometabolic outcomes included hypertension, type 2 diabetes, metabolic syndrome, and history of vascular disease. Multivariable regression models accounting for clustering at household and community levels and adjusted for age, sex, and their interaction were used to examine associations.

**Data availability statement:** All relevant data are within the paper and its Supporting Information files.

**Funding:** PJC; Universidad Internacional del Ecuador;EDM-INV-04-19; https://www.uide.edu.ec; The study funders had no role in study design, data collection and analysis, decision to publish, or preparation of the manuscript.

**Competing interests:** The authors have declared that no competing interests exist.

## Results

Elevated CRP (≥3 mg/L) was prevalent (50.9%) and significantly associated with hypertension (adjusted odds ratio [aOR] 1.69, 95% CI: 1.19–2.39), type 2 diabetes (aOR 2.53, 95% CI: 1.77–3.64), and metabolic syndrome (aOR 3.24, 95% CI: 2.30–4.58). Elevated CRP was also strongly linked to multimorbidity (aOR for ≥3–4 vs. no conditions: 5.96, 95% CI: 3.52–10.08, $P < 0.001$), as well as insulin resistance, low high-density lipoprotein, high triglycerides, and multiple adiposity measures. CRP associations were attenuated after adjusting for body mass index, suggesting adiposity as a mediator. In contrast, elevated total IgE (≥140 IU/mL) did not seem to be associated with cardiometabolic diseases. Total IgE levels were higher in men and associated with short stature, illiteracy, and obesity.

## Conclusions

Elevated CRP was strongly linked to cardiometabolic diseases and risk factors in this population, consistent with a model in which adiposity is a primary upstream driver of systemic inflammation. These findings highlight the importance of inflammation as a potential modifiable risk pathway and support the utility of CRP as a screening tool in low-resource transitional settings.

## Introduction

C-reactive protein (CRP) is an acute-phase protein produced primarily by the liver during acute inflammation triggered by infection or injury [1]. CRP levels rise rapidly but transiently to support immune defence and tissue repair processes. Repeated or unresolved inflammatory episodes can lead to chronic low-grade systemic inflammation (CSI) [2]. CSI is marked by modest but persistent elevations in CRP that contribute to endothelial dysfunction, insulin resistance, and atherosclerosis—key mechanisms considered to underlie the development of cardiometabolic diseases [2–5].

CSI has been linked to increased risk of cardiovascular diseases, including coronary artery disease and stroke, as well as hypertension (HTN) through effects of chronic inflammation on the vasculature [1,5,6]. CSI is thought also to contribute to the development of type 2 diabetes (T2D) by promoting insulin resistance and pancreatic beta-cell dysfunction [7,8] and has been strongly associated with the metabolic syndrome (MetS)—a constellation of cardiovascular disease risk factors including central obesity, dyslipidaemia, hypertension, and insulin resistance [9–11]. The detection of elevated levels of CRP using high sensitivity assays provide not only a useful tool for early risk detection of cardiometabolic conditions [1] but may also reflect a direct role of this inflammatory biomarker in their development [9,12].

Over the past 10–20 years, an epidemic of cardiometabolic diseases has emerged in many low and middle-income countries (LMICs) [13]. Cardiovascular diseases, stroke, HTN, and T2D and their complications are now the major causes of death in many Latin American countries including Ecuador where historically, infectious

diseases of childhood have predominated [14–15]. This epidemiological transition appears to have been driven by processes associated with rapid urbanization including lifestyle and environmental changes that elevate cardiometabolic risk. Urbanization is associated with increased consumption of processed, nutrient-poor but calorie-rich foods and reduced physical activity resulting in rising rates of obesity, and these changes may interact with early-life exposures such as poor childhood nutrition and recurrent infections [13,16]. Childhood undernutrition and infections can alter immune development, resulting in persistent inflammatory activation and metabolic dysregulation later in life [17–19]. The combination of these early insults with rapid lifestyle shifts, particularly in transitional rural and peri-urban communities, may accelerate the transition to CSI and premature cardiometabolic morbidity and mortality [20–22].

While CRP reflects innate immune activation and general inflammatory processes, total immunoglobulin E (IgE) results from type 2 T helper cell (Th2)-driven adaptive immune responses, where allergen or parasite exposures stimulate plasma cells to produce IgE of multiple specificities [23]. Mechanistically, Th2 inflammatory pathways including IgE-mediated inflammation could contribute to endothelial dysfunction, plaque instability, and insulin signalling impairment [24–27]. Studies from high-income countries (HICs) have shown that individuals with high serum IgE often have elevated blood pressure, dyslipidaemia, obesity, an increased risk of cardiometabolic diseases [28], and of cardiovascular mortality [29–31]. These findings suggest a potential link between allergic sensitization and cardiometabolic health [24–27,30].

In tropical rural populations, total IgE levels are typically elevated due to chronic exposure to helminths and other environmental pathogens, reflecting a regulated Th2 immune response rather than allergic sensitization [23,32,33]. This immunological profile is characterized by sustained activation of Th2 pathways alongside regulatory mechanisms that dampen pro-inflammatory pathways relevant to metabolic dysfunction [34,35]. As a result, elevated IgE in these settings may represent immune adaptation rather than pathology [36]. This distinct biological context, that differs from that observed in high-income settings, provides a strong rationale for examining total IgE in relation to cardiometabolic risk.

In this study, we examined the associations between two inflammatory biomarkers—CRP and total IgE— and cardiometabolic risk factors and diseases in adults from transitional rural communities in coastal Ecuador, where high infectious exposure coexists with a rapidly increasing burden of cardiometabolic disease. This setting provides a unique opportunity to contrast innate and type 2 inflammatory pathways in relation to metabolic dysfunction and better understand the role of inflammation in populations undergoing rapid epidemiological and nutritional transitions.

## Methods

### Study design, population, and setting

A detailed description of study methods is provided elsewhere [22]. This cross-sectional study aimed to describe the epidemiology and risk factors for cardiometabolic diseases in transitional rural communities and was conducted in ten adjacent rural agricultural communities within Abdon Calderón parish, district of Portoviejo, Manabí province, in a tropical region of coastal Ecuador. These communities are served primarily by a limited-capacity primary healthcare system, consisting of a local health centre providing basic outpatient services, with referral to a secondary-level hospital located approximately 30 km away. Access to diagnostic services for cardiometabolic diseases is limited with lack of capacity for routine screening for conditions such as hypertension and diabetes. Within this context, the present study was designed as a population-based epidemiological investigation, aiming to characterise the burden and distribution of cardiometabolic risk and its biological correlates in a setting where clinical detection is limited.

### Participants

The study population consisted primarily of Montubios, a marginalized Mestizo-derived ethnic group with distinct cultural and socioeconomic characteristics. These communities are characterized by ecological vulnerability and limited access to health services, located 10 km from a primary care facility and 30 km from the nearest regional hospital. Community representatives were involved in all stages of the study, including design, mobilization, and dissemination of results. Census

lists were compiled for each community, and all adults aged 18 years or older (with no upper age limit) were invited to participate through community assemblies. Exclusion criteria included refusal to provide written informed consent, current pregnancy, or childbirth within the previous three months. No exclusions were made on the basis of pre-existing chronic conditions (e.g., cancer or liver disease) or those that may influence IgE levels (e.g., asthma, allergic diseases, autoimmune conditions, or parasitic infections), as the study aimed to characterize biomarker distributions in a population-based sample. Participant recruitment and data collection took place from 31 July 2021 to 12 September 2021.

## Data and sample collection procedures

Data were collected following standardized procedures. Field staff underwent structured training prior to data collection, including standardization exercises for questionnaire administration and anthropometric measurements to ensure consistency across observers. Trained field staff administered structured and standardized questionnaires based on the WHO STEPwise approach to non-communicable chronic diseases (NCDs) risk factor surveillance (STEPS) to collect information on sociodemographic characteristics, lifestyle factors (e.g., smoking, alcohol use), occupational exposures, clinical history, and known diagnoses or treatment for NCDs [37]. Questionnaires were pilot-tested at the start of the study to ensure clarity and cultural appropriateness. Anthropometric measurements included weight (TANITA BF-60W), height (SECA 213), and waist and hip circumferences. Body composition was assessed using whole-body bioimpedance analysis (Bodystat 1500) to estimate fat mass. Anthropometric measurements were taken in duplicate using calibrated equipment, and the average of the two readings was used in analyses. Equipment was regularly checked and recalibrated according to manufacturer recommendations. Blood pressure was measured using an automated sphygmomanometer (OMRON 7051T) after 15 minutes of seated rest. If systolic blood pressure was ≥130 mmHg or diastolic ≥85 mmHg, a second reading was taken after a five-minute interval, and the average was used. While repeated measurements for all participants are recommended, this approach was adopted for feasibility in field conditions. Data were reviewed in the field for completeness and consistency, and electronic databases were subject to range and logic checks to identify potential data entry errors. Any discrepancies were resolved through verification against source records.

Fasting blood samples were collected in the early morning after an overnight fast. Capillary and venous samples were collected. Glycosylated haemoglobin (HbA1c) was measured in fresh blood using a fluorescent immunoassay (iChroma™, Boditech Med Inc., South Korea). Serum glucose, total cholesterol, high-density lipoprotein (HDL) cholesterol, and triglycerides were analysed using enzymatic colorimetric assays (Human Diagnostics, Germany). Insulin was measured using an immunoenzymatic assay (DIAsource ImmunoAssays, Belgium). Derived indices included the Homeostatic Model Assessment of Insulin Resistance (HOMA-IR) and the Visceral Adiposity Index (VAI) as described [22].

Laboratory analyses were conducted using standardized protocols. Assays were run in duplicate with suitable controls and against standard curves to ensure reliability of measurements. Total IgE was measured using a validated in-house assay as described previously [32]. High-sensitivity CRP (hs-CRP) was measured using a commercial assay (R&D systems) following the manufacturer's instructions.

## Disease definitions

Disease outcomes were defined using standard international criteria. T2D was defined as self-reported diagnosis or treatment, or HbA1c ≥ 6.5%. HTN was defined as self-reported treatment or systolic blood pressure (SBP) >=140 mm Hg or diastolic blood pressure (DBP) >=90 mm Hg on two measurements. MetS was diagnosed using the Harmonized criteria (≥3 out of 5 risk components) [38] that included: i) waist circumference of ≥94 cm in men and ≥88 cm in women [39]; ii) triglycerides ≥150 mg/dL or treatment with triglyceride-lowering drugs; iii) reduced HDL < 40 mg/dL in men and <50 mg/dL in women; iv) elevated blood pressure ≥130/85 mmHg or treatment with antihypertensives; and v) fasting glucose ≥100 mg/dL or treatment with glucose-lowering drugs. Adiposity indicators were defined as reported previously [22]: a) overweight (body mass index [BMI] ≥25) and obesity (BMI ≥ 30); b) abdominal obesity—waist circumference of ≥94 cm in men

and ≥88 cm in women [39]; c) elevated weight-to-height (WHtR)—≥ 0.5; d) body fat (free fat mass) was determined using bioimpedance [40] with increased body fat defined as ≥25% for men and ≥30% for women; e) visceral adiposity index (VAI) was calculated using a model of adipose distribution corrected for triglyceride and HDL levels [39] with high VAI according to age defined as—age < 30 –VAI > 2.52; age ≥ 30 & < 42 –VAI > 2.23; age ≥ 42 & < 52 –VAI > 1.92; age ≥ 52 & < 66 –VAI > 1.93; age ≥ 66 –VAI > 2. We defined elevated cholesterol as ≥200 mg/dL, elevated triglycerides as ≥150 mg/dL; and low HDL as <40 mg/dl in men and <50 mg/dl in women. Insulin resistance was defined by a Homeostatic Model Assessment (HOMA) index >2.5 [41,42]. A history of vascular disease was defined as a previous medical diagnosis of heart attack or stroke. Short stature was defined as height <165.5 cm in men and <153.3 cm in women [43]. Double burden of malnutrition (DBM) was defined as the co-occurrence short stature with overweight or obesity in the same individual [44]. Ethnicity was self-identified and categorised into two groups: mestizos and non-mestizos, the latter including Indigenous, Afro-Ecuadorians, Montubios, and White. Elevated CRP was defined as ≥3 mg/L [45]. Elevated total IgE was defined using a cut-off of 140 IU/mL [46], corresponding to the median value in this study population. This approach was adopted in the absence of standardized cut-offs for populations with high baseline IgE levels and is appropriate for exploratory analyses in this setting.

## Statistical analysis

We explored the associations between CRP or total IgE and presence of chronic diseases (T2D, HTN, MetS), and history of vascular events (stroke or myocardial infarction) or presence of multiple comorbidities, and a variety of potential risk factors including cardiometabolic risk factors such as blood lipids, indicators of adiposity, systolic and diastolic blood pressure, fasting blood glucose, HbA1c, and HOMA. Because of the three-level hierarchical structure of the data (individuals nested in households and communities), multilevel regression techniques tailored to the nature of the statistical outcomes such as regression (for continuous responses) or logistic regression (for binary responses) were used to infer standard errors and p-values. All analyses were exploratory in nature and were minimally adjusted for age, sex, and age-sex interaction (when significant), which are key determinants of both inflammatory markers and cardiometabolic outcomes. Age was modelled as a continuous variable across the full adult age range (≥18 years), including a quadratic term to account for non-linearity. A parsimonious adjustment strategy was adopted to avoid overadjustment, particularly for variables such as body mass index (BMI), which may lie on the causal pathway between systemic inflammation and cardiometabolic disease. In addition, age and sex data from the Census population in district of Portoviejo [22] were used to derive post-stratification weights to inflate misrepresented groups and reduce the effect of overrepresented groups in the sample [47]. These techniques allowed the interpretation of the findings for CRP and total IgE as their corresponding age-specific levels and age-specific prevalence in the population. Analyses were also performed to assess the role of different exposures as potential mediating factors in causal pathways for disease outcomes within a conceptual framework–using this approach, controlling for an intermediate factor in the causal mechanism would be expected to decrease the estimate of effect for the disease outcomes towards 1 [48]. The hierarchical modelling framework was used to explore potential pathways linking inflammation, adiposity, and cardiometabolic outcomes, rather than to estimate fully adjusted causal effects. Data analyses were performed using Stata version 18 (StataCorp, College Station, Texas, USA).

## Ethics statement

Informed written consent was obtained from each participant, and the study protocol was approved by Bioethics Committee of the UTE University (Comite de Etica de Investigacion en Seres Humanos de la Universidad UTE, CEISH UTE 2019-1121-03 and 094-CEISH-jcm). All procedures involving human participants were conducted in accordance with the ethical standards of the institutional and national research committees and with the 1964 Helsinki Declaration and its later amendments.

## Results

### Study population and characteristics

A total of 931 adults aged 18 years and older from 10 communities were included in the analysis (Fig 1). Sociodemographic factors, risk factors and adiposity indicators, and biochemical risk factors are shown in Table 1.

   Mean age was 44.7 years (standard deviation 18.1) and 42.5% were male. Most participants lived with a partner (66.1%); 24.3% were functionally illiterate; and 48.3% self-identified as non-mestizo. Most men (62.6%) worked in agriculture while most women (80.9%) were housewives. Men (54.8%) were more likely than women (19.6%) to do physically demanding work. Cardiometabolic conditions were highly prevalent in this population: HTN affected 44.7% (women 46.2% vs. men 42.7%), T2D affected 26.3% (women 29.5% vs. men 22.0%), MetS affected 58.0% (women 60.9% vs. men 54.0%), and history of vascular events was reported by 8.7% (women 7.9% vs. men 9.9%). There was evidence that vascular events occurred at relatively young ages in both sexes: 5% of those aged below 40 years reported such a history (S1 Fig). There was considerable overlap between the 4 chronic diseases syndrome (Fig 2), and relatively few individuals (11%) had T2D and/or HTN but not the MetS. A history of vascular disease (used to indicate advanced atherosclerotic disease) overlapped almost completely with the other chronic conditions particularly the MetS. Multimorbidity, determined by the co-occurrence of these conditions, was frequent with 43.2% (women 45.6% vs. men 39.9%) having 2 or more of the 4 chronic conditions. Short stature was present in 55.4% of participants (women 56.1% vs. men 54.6%), overweight or obesity in 65.4% (women 69.5% vs. men 59.8%) and both representing the double burden of malnutrition in 37.6% (women 41.1% vs. men 32.8%). Indicators of adiposity, insulin resistance, and elevated lipids were more frequent in women (insulin resistance [women 69.5% vs. men 43.9%; abdominal obesity [women 66.4% vs. men 49.5%]; elevated cholesterol [women 50.5% vs. men 44.2%]). Overall, these data indicate a high burden of cardiometabolic risk and multimorbidity in this population, providing an important context for interpreting differences in inflammatory markers across demographic and clinical groups.

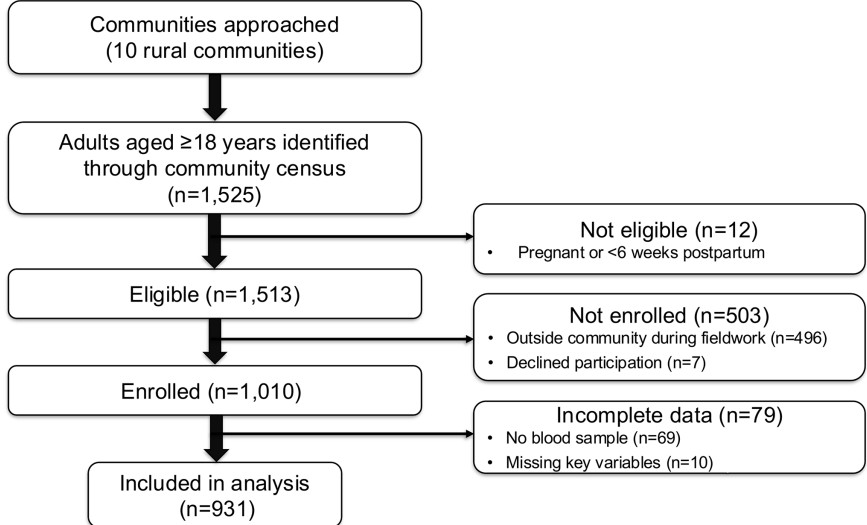

**Fig 1. Flow diagram of participant recruitment, eligibility, and inclusion in the analytical sample (n = 931).** All adults aged ≥18 years from 10 rural communities were identified through census enumeration and invited to participate. After accounting for exclusions and incomplete data, a total of 931 individuals were included in the final analysis. Numbers at each stage are shown.

**Table 1. Baseline characteristics and cardiometabolic risk factors and diseases among 931 adults in rural coastal Ecuador, stratified by sex.**

| Variable | Category | Total (n = 931) n (%) | Women (n = 535) n (%) | Men (n = 396) n (%) |
|---|---|---|---|---|
| **Age distribution and risk factors** | | | | |
| Age group (years) | 18-29 | 229 (24.6) | 131 (24.5) | 98 (24.8) |
| | 30-39 | 158 (17.0) | 100 (18.7) | 58 (14.6) |
| | 40-49 | 183 (19.7) | 107 (20.0) | 76 (19.2) |
| | 50-59 | 149 (16.0) | 82 (15.3) | 67 (16.9) |
| | 60-69 | 104 (11.2) | 56 (10.5) | 48 (12.1) |
| | ≥ 70 | 108 (11.5) | 59 (11.0) | 49 (12.4) |
| Lives with partner | Yes | 615 (66.1) | 362 (67.7) | 253 (63.9) |
| Functional illiteracy | Yes | 226 (24.3) | 114 (21.3) | 112 (28.3) |
| Ethnicity | Non-mestizo | 450 (48.3) | 229 (42.9) | 221 (55.8) |
| | Mestizo | 481 (51.7) | 306 (57.2) | 175 (44.2) |
| Occupation | Agricultural worker | 261 (28.1) | 13 (2.4) | 248 (62.6) |
| | Household chores | 436 (46.8) | 433 (80.9) | 3 (0.8) |
| | Non-agricultural workers | 234 (25.1) | 89 (16.6) | 145 (36.6) |
| Recent alcohol consumption | Yes | 575 (62.2) | 270 (50.8) | 305 (77.6) |
| Current smoker | Yes | 72 (7.7) | 4 (0.8) | 68 (17.2) |
| Physically-demanding work | Yes | 319 (34.6) | 104 (19.6) | 215 (54.8) |
| Contact with agrochemicals | Yes | 253 (27.2) | 47 (8.8) | 206 (52.0) |
| **Chronic diseases** | | | | |
| Hypertension | Yes | 416 (44.7%) | 247 (46.2%) | 169 (42.7%) |
| Diabetes | Yes | 245 (26.3%) | 158 (29.5%) | 87 (22.0%) |
| Metabolic syndrome | Yes | 540 (58.0%) | 326 (60.9%) | 214 (54.0%) |
| Vascular events | Yes | 81 (8.7%) | 42 (7.9%) | 39 (9.9%) |
| Multimorbidity | No disease | 287 (30.8%) | 153 (28.6%) | 134 (33.8%) |
| | One disease only | 242 (26.0%) | 138 (25.8%) | 104 (26.3%) |
| | Two diseases only | 247 (26.5%) | 139 (26.0%) | 108 (27.3%) |
| | Three to four diseases | 155 (16.7%) | 105 (19.6%) | 50 (12.6%) |
| **Nutritional and biochemical indicators** | | | | |
| Short stature | men < 165.5 cm; women<153.3 cm | 516 (55.4%) | 300 (56.1%) | 216 (54.6%) |
| Obesity | Not overweight (BMI < 25) | 322 (34.6%) | 164 (30.5%) | 159 (40.2%) |
| | Overweight (BMI 25–29) | 367 (39.4%) | 212 (39.6%) | 155 (39.1%) |
| | Obese (BMI ≥ 30) | 242 (26.0%) | 160 (29.9%) | 82 (20.7%) |
| Double burden of malnutrition | None | 156 (16.8%) | 83 (15.5%) | 73 (18.4%) |
| | Short stature only | 166 (17.8%) | 80 (14.9%) | 86 (21.7%) |
| | Overweight/obesity only | 259 (27.8%) | 152 (28.4%) | 107 (27.0%) |
| | Short stature AND overweight/ obesity | 350 (37.6%) | 220 (41.1%) | 130 (32.8%) |
| Abdominal obesity | men ≥ 94 cm; women ≥88 cm | 551 (59.2) | 355 (66.4) | 196 (49.5) |
| Elevated WHtR | ≥0.5 | 791 (85.0) | 475 (88.8) | 316 (79.8) |
| Elevated body fat | men (≥25%); women (≥30%) | 696 (95.9) | 412 (97.9) | 284 (93.1) |
| Elevated Visceral Adiposity Index | See legend | 571 (61.3) | 368 (68.8) | 203 (51.3) |
| Elevated cholesterol | ≥200 mg/dl | 445 (47.8) | 270 (50.5) | 175(44.2) |
| Elevated triglycerides | ≥150 mg/dL | 434 (44.6) | 257 (48.0) | 177 (44.7) |
| Low HDL | Men < 40 mg/dl; women<50 mg/dL | 524 (56.3) | 367 (68.6) | 157 (39.7) |
| Insulin resistance | HOMA-IR > 2.5 | 546 (58.6) | 372 (69.5) | 174 (43.9) |

Values represent unweighted frequencies (%) unless otherwise indicated. Age range: 18 years and older (no upper age limit). Functional illiteracy was defined as ≤3 years of formal education [29]. Abbreviations: BMI – body mass index; WHtR – waist-to-height ratio; VAI – visceral adiposity index; HDL – high-density lipoprotein; HOMA-IR – Homeostatic Model Assessment of Insulin Resistance. High VAI was defined using age-specific cut-offs as described in Methods. Missing data: physically demanding work (n = 8). Denominators vary slightly due to missing data. This table describes the high burden of cardiometabolic risk factors and multimorbidity in the study population.

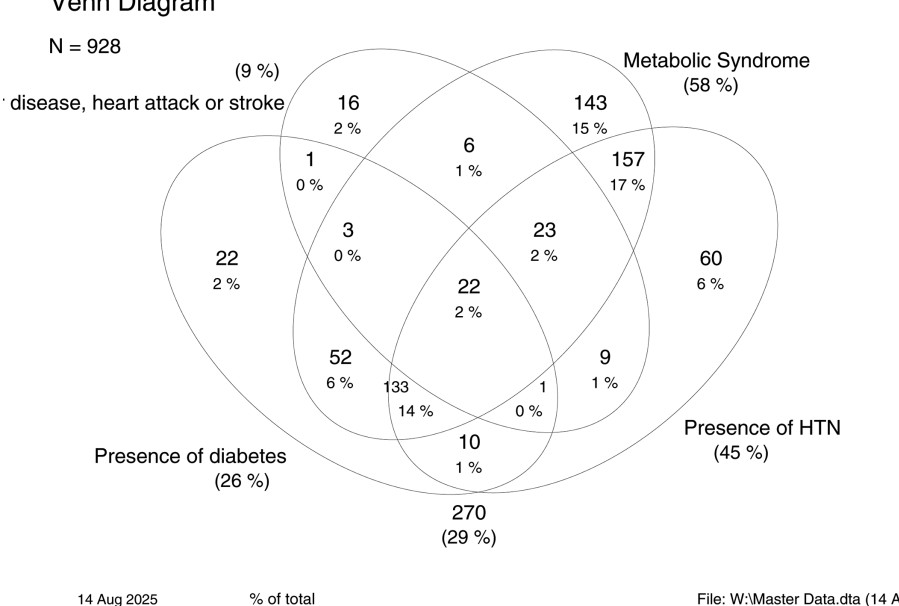

**Venn Diagram**

N = 928

disease, heart attack or stroke (9 %)

Metabolic Syndrome (58 %)

16 — 2 %

143 — 15 %

6 — 1 %

1 — 0 %

157 — 17 %

22 — 2 %

3 — 0 %

23 — 2 %

60 — 6 %

22 — 2 %

52 — 6 %

133 — 14 %

1 — 0 %

9 — 1 %

Presence of diabetes (26 %)

10 — 1 %

Presence of HTN (45 %)

270 (29 %)

14 Aug 2025      % of total      File: W:\Master Data.dta (14 Aug 2025 )

**Fig 2. Overlap of cardiometabolic conditions among 928 adults in rural coastal Ecuador.** Venn diagram showing the co-occurrence of hypertension (HTN), type 2 diabetes (T2D), metabolic syndrome (MetS), and history of vascular events (stroke or myocardial infarction) among 928 participants. Values represent the number of individuals within each overlap category. A total of 270 individuals (29%) had none of these conditions. Data are unweighted and illustrate the high burden of multimorbidity and clustering of cardiometabolic diseases in the study population.

## Associations of potential risk factors and cardiometabolic diseases with C-reactive protein (CRP)

The prevalence of elevated levels of CRP (>=3 mg/L) was high across all ages (50.9%). Age-specific patterns were non-linear, with levels increasing into later adulthood and peaking around 60 years. Elevated CRP was consistently more frequent in women than men across the age range (Fig 3A and S2 Fig). Associations between potential risk exposures, chronic diseases, or indicators of adiposity and biochemical risk factors and CRP analysed either as binary or continuous, are shown in Table 2. Elevated CRP levels showed a clear relationship with increasing cardiometabolic burden, in contrast to patterns observed for total IgE (see below).

When categorized as a binary variable (i.e., elevated vs. normal levels of CRP), CRP was significantly associated with age (per additional year, OR 1.02, 95% CI 1.01–1.03), female sex (males vs. females, OR 0.25, 95% CI 0.17–0.38), and the interaction between ethnicity and sex (interaction P = 0.033). Elevated CRP was significantly associated with having HTN (adj. OR 1.69, 95% CI 1.19–2.39), T2D (adj. OR 2.53, 95% CI 1.77–3.64), and the MetS (adj. OR 3.24, 95% CI 2.30–4.58), but not with a history of vascular disease. These associations were consistent across multiple cardiometabolic outcomes and strengthened with increasing multimorbidity, indicating a robust relationship between CRP and disease burden. There was evidence for stronger associations between elevated CRP with greater number of morbid conditions (for example, 3–4 vs. 0 conditions, adj. OR 5.96, 95% CI 3.52–10.08). Prevalence of elevated CRP was greater with greater number of chronic diseases across ages (Fig 4A and 4B), although prevalence declined with age, particularly in females (Fig 4A). In contrast, prevalence of elevated CRP increased with age in males and females (Fig 4A and 4B). Elevated CRP was only associated with nutritional status in the presence of overweight/obesity and was not observed among those with short stature without overweight/obesity (Table 2). There were strong significant associations between cardiometabolic risk factors and elevated CRP including insulin resistance, elevated triglycerides (but not cholesterol), low HDL, and all indicators of adiposity (i.e., elevated WhtR, elevated body fat, high VAI, and abdominal obesity) (Table 2). Similar

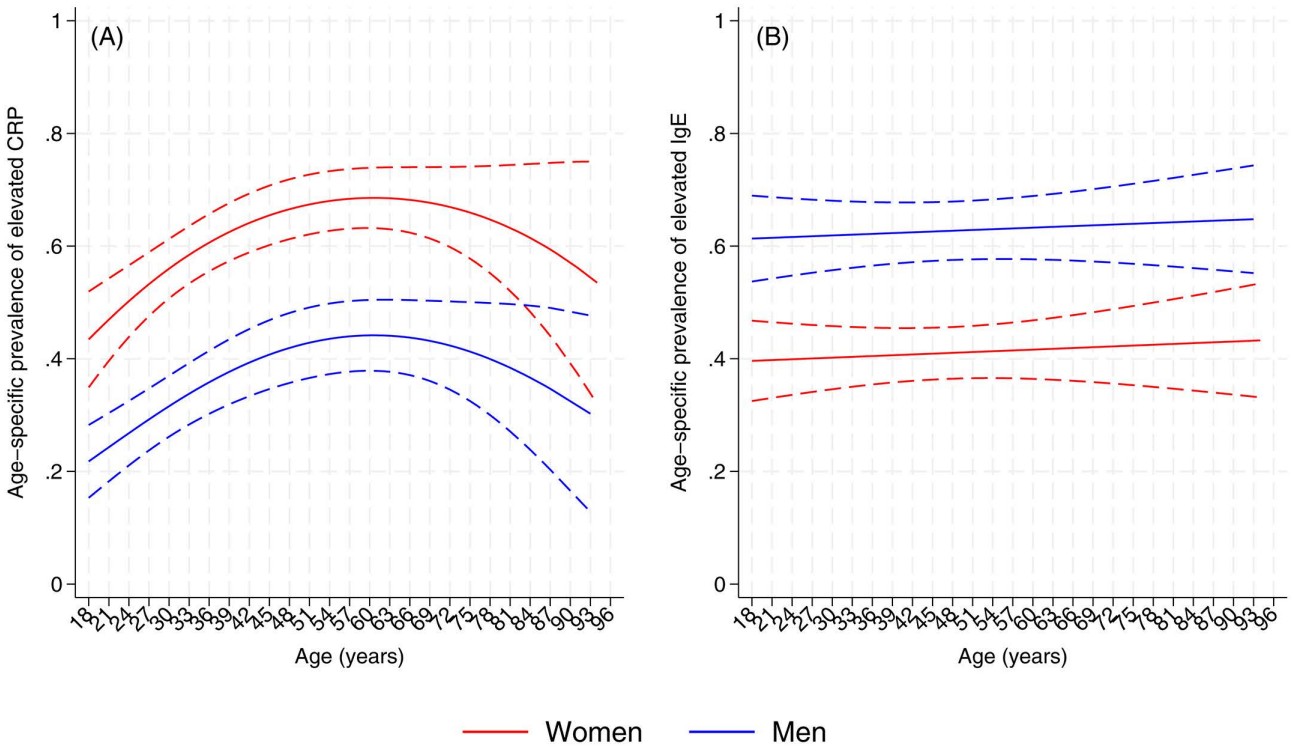

**Fig 3. Age- and sex-specific predicted prevalence of elevated inflammatory biomarkers.** Population-representative predicted prevalence of elevated C-reactive protein (CRP ≥ 3 mg/L) (A) and elevated total immunoglobulin E (IgE ≥ 140 IU/mL) (B) across age, stratified by sex. Population-weighted multilevel logistic regression models were used. Dotted lines represent 95% confidence intervals. Analyses include all adults aged ≥18 years. These plots illustrate distinct age- and sex-specific patterns for CRP and IgE across the adult life course.

associations were observed when CRP was analysed as a continuous variable (Table 2). Taken together, CRP showed a consistent pattern of association across adiposity, metabolic risk factors, and clinical outcomes, distinguishing it from total IgE, which did not demonstrate comparable relationships.

### Associations of potential risk factors or cardiometabolic diseases with total IgE

In contrast to CRP, total IgE showed a markedly different pattern of association with cardiometabolic risk factors and diseases. Levels of total IgE were high in this population with 50% having levels greater 140 IU/mL (median used as median cutoff; interquartile range (Q1-Q3, 56.2–399.7). Changes in proportions of participants with elevated total IgE (i.e., >=140 IU/mL) with increasing age for men and women are shown in Fig 3B, while changes in concentrations are shown in S2 Fig. Associations between potential risk exposures, chronic diseases, or indicators of adiposity and biochemical risk factors and total IgE analysed either as binary or continuous, are shown in Table 3. Elevated total IgE did not exhibit a significant association with age but was observed more frequently in men (vs. women, OR 2.41, 95% CI 1.82–3.19). In sex-adjusted analyses, elevated total IgE was associated with functional illiteracy (adj. OR 1.54, 95% CI 1.09–2.17), short stature (adj. OR 1.61, 95% CI 1.19–2.17), obesity (vs. not overweight, adj. OR 1.64, 95% CI 1.10–2.45) and elevated body fat (adj. OR 2.64, 95% CI 1.14–6.12) but other indicators of nutrition, adiposity or biochemical risk factors did not show statistically significant associations. Elevated total IgE was not significantly associated with any of the chronic disease outcomes measured or multimorbidity. This contrasts sharply with CRP, which showed strong and consistent associations with these outcomes. Similar findings were observed when total IgE was analysed as a continuous variable. There

**Table 2. Adjusted associations between C-reactive protein (CRP) and cardiometabolic risk factors, chronic conditions, and multimorbidity.**

| Variables | Categories | CRP | | CRP (<3 vs. ≥3 mg/L) | | | CRP (continuous) | | |
|---|---|---|---|---|---|---|---|---|---|
| | | <3 mg/L (n=457) | ≥3 mg/L (n=474) | OR | 95% CI | p-value | Fold-change | 95% CI | p-value |
| **Demographics** | | | | | | | | | |
| Age | continuous | | | **1.02** | **1.01-1.03** | **<0.001** | **1.01** | **1.00-1.01** | **<0.001** |
| Age$^2$ | continuous | | | **0.9994** | **0.9990-0.9998** | **0.006** | **0.9997** | **0.9995-0.9999** | **<0.001** |
| Sex | Female | 214(40.0%) | 321(60.0%) | 1 | | | 1 | | |
| | Male | 243(61.4%) | 153(38.6%) | **0.25** | **0.17-0.38** | **<0.001** | **0.61** | **0.53-0.71** | **<0.001** |
| Ethnicity | Non-Mestizo | 211(46.9%) | 239(53.1%) | 1 | | | 1 | | |
| | Mestizo | 246(51.1%) | 235(48.9%) | **0.61** | **0.41-0.90** | **0.012** | 0.97 | 0.83-0.14 | 0.709 |
| Interaction: Ethnicity × Sex | | | | **1.90** | **1.05-3.43** | **0.033** | | | |
| **Risk Exposures** | | | | | | | | | |
| Lives with partner | No | 178(56.3%) | 138(43.7%) | 1 | | | 1 | | |
| | Yes | 279(45.4%) | 336(54.6%) | **1.44** | **1.04-2.01** | **0.029** | 1.05 | 0.89-1.24 | 0.564 |
| Functional illiteracy | No | 356(50.5%) | 349(49.5%) | 11 | | | 1 | | |
| | Yes | 101(44.7%) | 125(55.3%) | 1.11 | 0.75-1.64 | 0.594 | 1.10 | 0.88-1.36 | 0.411 |
| Occupation | Agricultural worker | 157(60.2%) | 104(39.8%) | 1 | | | 1 | | |
| | Household chores | 166(38.1%) | 270(61.9%) | 1.40 | 0.78-2.52 | 0.254 | 1.25 | 0.91-1.71 | 0.175 |
| | Non-agricultural workers | 134(57.3%) | 100(42.7%) | 1.15 | 0.73-1.82 | 0.548 | 1.17 | 0.93-1.48 | 0.168 |
| Recent alcohol consumption | No | 163(46.7%) | 186(53.3%) | 1 | | | 1 | | |
| | Yes | 290(50.4%) | 285(49.6%) | 1.22 | 0.90-1.68 | 0.204 | 1.06 | 0.897-1.251 | 0.494 |
| Current smoker | No | 411(47.9%) | 447(52.1%) | 1 | | | 1 | | |
| | Yes | 45(62.5%) | 27(37.5%) | 0.83 | 0.46-1.50 | 0.530 | 0.88 | 0.653-1.176 | 0.377 |
| Physically demanding work | No | 275(45.5%) | 329(54.5%) | 1 | | | 1 | | |
| | Yes | 178(55.8%) | 141(44.2%) | 0.83 | 0.60-1.14 | 0.243 | 0.92 | 0.79-1.08 | 0.306 |
| Contact with agrochemicals | No | 314(46.3%) | 364(53.7%) | 1 | | | 1 | | |
| | Yes | 143(56.5%) | 110(43.5%) | 1.00 | 0.69-1.44 | 0.981 | 0.97 | 0.81-1.17 | 0.756 |
| **Chronic Diseases** | | | | | | | | | |
| Hypertension | No | 289(56.1%) | 226(43.9%) | 1 | | | 1 | | |
| | Yes | 168(40.4%) | 248(59.6%) | **1.69** | **1.19-2.39** | **0.003** | **1.27** | **1.05-1.54** | **0.012** |
| Diabetes | No | 379(55.2%) | 307(44.8%) | 1 | | | 1 | | |
| | Yes | 78(31.8%) | 167(68.2%) | **2.53** | **1.77-3.64** | **<0.001** | **1.59** | **1.32-1.91** | **<0.001** |
| Metabolic syndrome | No | 257(65.7%) | 134(34.3%) | 1 | | | 1 | | |
| | Yes | 200(37.0%) | 340(63.0%) | **3.24** | **2.30-4.58** | **<0.001** | **1.79** | **1.51-2.12** | **<0.001** |
| Vascular events | No | 419(49.5%) | 428(50.5%) | 1 | | | 1 | | |
| | Yes | 35(43.2%) | 46(56.8%) | 1.43 | 0.84-2.44 | 0.187 | 1.042 | 0.85-1.28 | 0.697 |
| Multimorbidity | 0 diseases | 229(46.3%) | 113(22.3%) | 1 | | | 1 | | |
| | 1 disease | 115(23.2%) | 127(25.1%) | **2.53** | **1.63-3.93** | **<0.001** | **1.56** | **1.27-1.92** | **<0.001** |
| | 2 diseases | 93(18.8%) | 143(28.2%) | **4.28** | **2.69-6.82** | **<0.001** | **1.90** | **1.54-2.34** | **<0.001** |
| | 3-4 diseases | 58(11.7%) | 124(24.5%) | **5.96** | **3.52-10.08** | **<0.001** | **2.35** | **1.80-3.06** | **<0.001** |
| **Indicators of adiposity and biochemical risk factors** | | | | | | | | | |
| Height | Continuous | | | 1.00 | 0.98-2.03 | 0.927 | 1.05 | 0.89-1.24 | 0.564 |
| Short stature | No | 253(51.1%) | 233(46.0%) | 1 | | | 1 | | |

*(Continued)*

| Variables | Categories | CRP | | CRP (<3 vs. ≥ 3 mg/L) | | | CRP (continuous) | | |
|---|---|---|---|---|---|---|---|---|---|
| | Yes | 242(48.9%) | 274(54.0%) | 1.22 | 0.89-1.67 | 0.214 | 1.14 | 0.97-1.35 | 0.115 |
| BMI | Continuous | | | **1.18** | **1.14-1.22** | **<0.001** | **1.09** | **1.08-1.11** | **<0.001** |
| BMI | Not overweight | 226(70.2%) | 96(29.8%) | 1 | | | 1 | | |
| | Overweight | 163(44.4%) | 204(55.6%) | **2.70** | **1.90-3.85** | **<0.001** | **1.73** | **1.48-2.02** | **<0.001** |
| | Obese | 68(28.1%) | 174(71.9%) | **6.26** | **4.14-9.46** | **<0.001** | **2.68** | **2.20-3.27** | **<0.001** |
| Double burden of malnutrition | None | 112(24.5%) | 44(9.3%) | 1 | | | 1 | | |
| | Short stature only | 114(25.0%) | 52(11.0%) | **1.21** | **0.71-2.08** | 0.478 | 1.18 | 0.94 −1.48 | 0.157 |
| | Overweight/ Obese only | 103(22.5%) | 156(32.9%) | **3.76** | **2.30-6.14** | **<0.001** | **2.13** | **1.73-2.64** | **<0.001** |
| | Both | 128(28.0%) | 222(46.8%) | **4.41** | **2.73-7.14** | **<0.001** | **2.33** | **1.85-2.92** | **<0.001** |
| Insulin resistance | No | 247(64.2%) | 138(35.8%) | 1 | | | 1 | | |
| | Yes | 210(38.5%) | 336(61.5%) | **2.45** | **1.82-3.29** | **<0.001** | **1.59** | **1.35-1.88** | **<0.001** |
| WHtR elevated | No | 113(80.7%) | 27(19.3%) | 1 | | | 1 | | |
| | Yes | 344(43.5%) | 447(56.5%) | **4.81** | **2.85-8.11** | **<0.001** | **2.31** | **1.86-2.86** | **<0.001** |
| Elevated body fat | No | 25(83.3%) | 5(16.7%) | 1 | | | 1 | | |
| | Yes | 333(47.8%) | 363(52.2%) | **4.47** | **1.47-13.61** | **0.008** | **1.75** | **1.32-2.31** | **<0.001** |
| High VAI | No | 228(63.3%) | 132(36.7%) | 1 | | | 1 | | |
| | Yes | 229(40.1%) | 342(59.9%) | **2.27** | **1.59-3.22** | **<0.001** | **1.43** | **1.19-1.72** | **<0.001** |
| Abdominal obesity | No | 267(70.3%) | 113(29.7%) | 1 | | | 1 | | |
| | Yes | 190(34.5%) | 361(65.5%) | **3.75** | **2.69-5.24** | **<0.001** | **2.00** | **1.70-2.35** | **<0.001** |
| Elevated cholesterol | No | 217(54.9%) | 178(45.1%) | 1 | | | 1 | | |
| | Yes | 240(44.8%) | 296(55.2%) | 1.25 | 0.93-1.70 | 0.146 | 1.13 | 0.96-1.33 | 0.148 |
| Elevated triglycerides | No | 278(55.9%) | 219(44.1%) | 1 | | | 1 | | |
| | Yes | 179(41.2%) | 255(58.8%) | **1.61** | **1.20-2.16** | **0.002** | **1.20** | **1.03-1.39** | **0.020** |
| Low HDL | No | 237(58.2%) | 170(41.8%) | 1 | | | 1 | | |
| | Yes | 220(42.0%) | 304(58.0%) | **1.76** | **1.28-2.42** | **0.001** | **1.40** | **1.20-1.64** | **<0.001** |

Associations are presented for CRP analysed as a binary outcome (<3 vs. ≥ 3 mg/L; odds ratios [ORs]) and as a continuous outcome (geometric mean ratios [GMRs]). Estimates were derived from population-weighted multilevel regression models adjusted for age, age², sex, and age–sex interaction (where significant), with clustering at household and community levels. ORs represent the adjusted odds of elevated CRP (≥3 mg/L) per unit increase in the explanatory variable. GMRs represent the fold change in CRP concentration. Analyses include all adults aged ≥18 years. Abbreviations: BMI – body mass index; WHtR – waist-to-height ratio; VAI – visceral adiposity index; HDL – high-density lipoprotein. These results demonstrate strong and consistent associations between CRP and cardiometabolic risk factors, diseases, and multimorbidity.

was evidence that those with the double burden of malnutrition had greater levels of total IgE compared to those with normal anthropometry (fold change, adj. OR 2.00, 95% CI 1.29–3.10). Overall, while CRP tracked closely with cardiometabolic risk and disease burden, total IgE appeared to reflect environmental and socioeconomic exposures rather than clinical cardiometabolic outcomes.

## Hierarchical analysis for associations between CRP and key cardiometabolic indicators

CRP or BMI may be intermediate or proximal factors in associations between elevated CRP/BMI and cardiometabolic diseases. To explore these relationships, we analysed the effects of CRP and/or BMI on key cardiometabolic indicators for these diseases: systolic and diastolic blood pressure for HTN and glycosylated haemoglobin and insulin resistance for T2D (Table 4). Increased CRP and BMI (classified as overweight or obese) were strongly associated with all 5 cardiometabolic indicators. When controlling for both CRP and BMI in these models, there was evidence of a reduction in

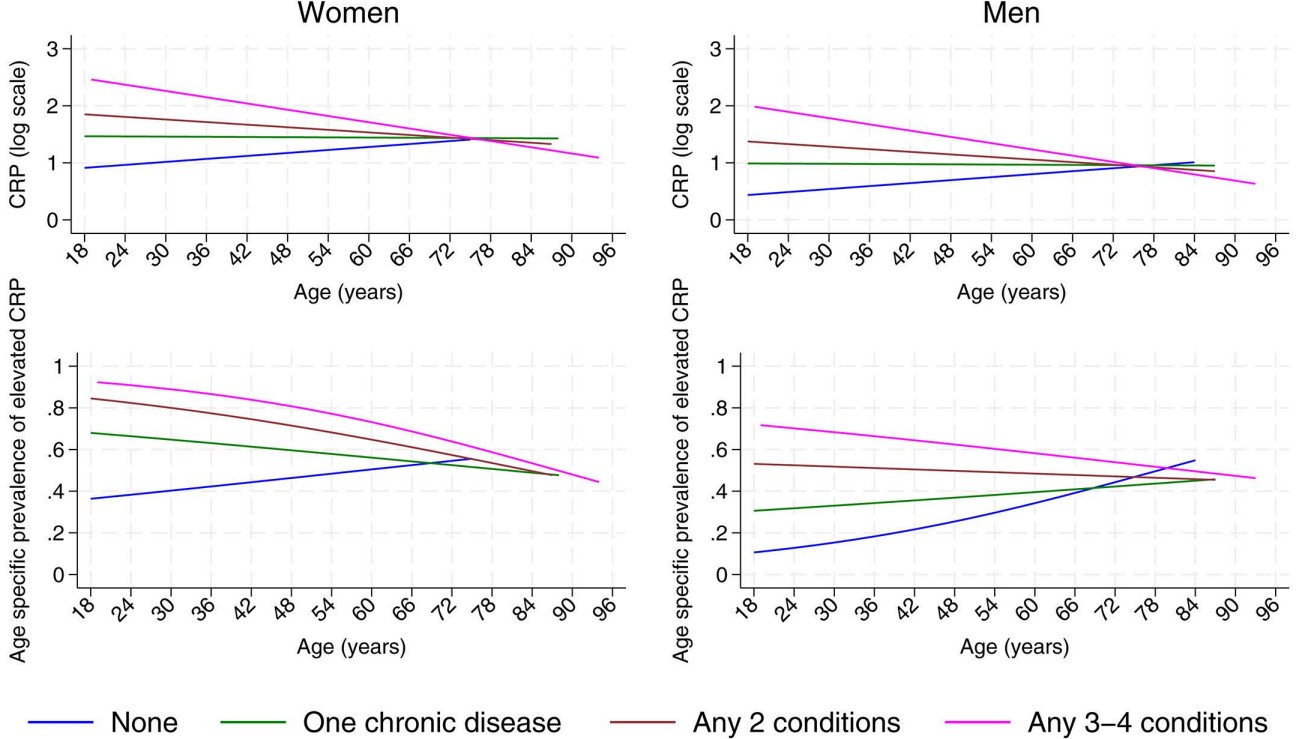

**Fig 4. Age-specific patterns of CRP by sex and multimorbidity status.** Population-representative predicted CRP levels (log-transformed) and prevalence of elevated CRP (≥3 mg/L) across age, stratified by sex and number of cardiometabolic conditions (0, 1, 2, or ≥3 conditions). Population weighted multilevel regression techniques tailored to the nature of the outcomes (continuous or binary) were used. These plots illustrate how systemic inflammation varies with age, sex, and increasing multimorbidity burden.

the estimates of effect for CRP but not BMI when considering associations with systolic and diastolic blood pressure. In the case of HbA1c and insulin resistance (measured using HOMA-IR), controlling for BMI did not seem to translate into an effect on the association of CRP with HbA1c but did substantially reduce the magnitude of the association with insulin resistance. Controlling for CRP did not seem to change the BMI effect on the two metabolic indicators in this population. These findings suggest that adiposity may lie upstream of CRP in pathways linking inflammation to cardiometabolic risk, further reinforcing the central role of obesity in shaping inflammatory and metabolic profiles in this population.

In summary, the two inflammatory markers showed clearly divergent patterns. CRP was strongly and consistently associated with cardiometabolic risk factors, diseases, and multimorbidity, whereas total IgE—despite its high prevalence— was not associated with these outcomes and instead likely reflected underlying social and environmental gradients.

## Discussion

In the present analysis, we used data from a cross-sectional study of adults living in transitional rural communities in coastal Ecuador to explore relationships between markers of systemic inflammation—hs-CRP and total IgE—and the presence of cardiometabolic risk factors and diseases. In these communities, a high proportion of adults had cardiometabolic risk factors and diseases, representing a population at high risk of premature morbidity and mortality [22]. We observed a high prevalence of the two indicators of inflammation used, CRP and total IgE – over 50% had elevated levels of one or the other indicator. Strong associations were observed between elevated CRP and the presence of cardiometabolic diseases and multimorbidity, insulin resistance and indicators of adiposity. However, overweight or obesity appeared

**Table 3. Adjusted associations between total immunoglobulin E (IgE) and cardiometabolic risk factors, chronic conditions, and multimorbidity.**

| Variables | Categories | IgE | | IgE (<vs. ≥ median) | | | IgE (continuous on log scale) | | |
|---|---|---|---|---|---|---|---|---|---|
| | | <140 IU/mL (n = 469) | ≥140 IU/mL (n = 462) | OR | 95% CI | p-value | Fold-change | 95% CI | p-value |
| **Demographics** | | | | | | | | | |
| Age | continuous | | | 1.00 | 0.99-1.01 | 0.765 | 1.00 | 1.00-1.01 | 0.380 |
| Sex | Female | 318(59.4%) | 217(40.6%) | 1 | | | 1 | | |
| | Male | 151(38.1%) | 245(61.9%) | **2.41** | **1.82-3.19** | **<0.001** | **1.98** | **1.62-2.42** | **<0.001** |
| Ethnicity | Non-mestizo | 229(50.9%) | 221(49.1%) | 1 | | | 1 | | |
| | Mestizo | 240(49.9%) | 241(50.1%) | 1.04 | 0.78-1.40 | 0.779 | 0.92 | 0.75-1.14 | 0.456 |
| **Risk Exposures** | | | | | | | | | |
| Lives with partner | No | 147(46.5%) | 169(53.5%) | 1 | | | 1 | | |
| | Yes | 322(52.4%) | 293(47.6%) | 0.87 | 0.65-1.17 | 0.350 | **0.79** | **0.64-0.97** | **0.024** |
| Functional illiteracy | No | 375(53.2%) | 330(46.8%) | 1 | | | 1 | | |
| | Yes | 94(41.6%) | 132(58.4%) | **1.54** | **1.09-2.17** | **0.015** | **1.58** | **1.27-1.97** | **<0.001** |
| Occupation | Agricultural worker | 96(36.8%) | 165(63.2%) | 1 | | | 1 | | |
| | Household chores | 262(60.1%) | 174(39.9%) | 0.68 | 0.39-1.17 | 0.159 | 0.78 | 0.53-1.13 | 0.182 |
| | Non-agricultural workers | 111(47.4%) | 123(52.6%) | 0.66 | 0.43-1.02 | 0.058 | 0.74 | 0.54-1.01 | 0.056 |
| Recent alcohol consumption | No | 189(54.2%) | 160(45.8%) | 1 | | | 1 | | |
| | Yes | 273(47.5%) | 302(52.5%) | 1.03 | 0.70-1.52 | 0.874 | 0.94 | 0.77-1.15 | 0.573 |
| Current smoker | No | 445(51.9%) | 413(48.1%) | 1 | | | 1 | | |
| | Yes | 23(31.9%) | 49(68.1%) | 1.50 | 0.84-2.69 | 0.169 | 1.15 | 0.66-2.00 | 0.618 |
| Physically demanding work | No | 325(53.8%) | 279(46.2%) | 1 | | | 1 | | |
| | Yes | 138(43.3%) | 181(56.7%) | 1.13 | 0.81-1.57 | 0.481 | 0.96 | 0.77-1.20 | 0.736 |
| Contact with agrochemicals | No | 360(53.1%) | 318(46.9%) | 1 | | | 1 | | |
| | Yes | 109(43.1%) | 144(56.9%) | 0.94 | 0.64-1.39 | 0.757 | 1.05 | 0.80-1.37 | 0.743 |
| **Chronic Conditions** | | | | | | | | | |
| Hypertension | No | 257(49.9%) | 258(50.1%) | 1 | | | 1 | | |
| | Yes | 212(51.0%) | 204(49.0%) | 0.94 | 0.71-1.26 | 0.697 | 1.01 | 0.84-1.22 | 0.907 |
| Diabetes | No | 342(49.9%) | 344(50.1%) | 1 | | | 1 | | |
| | Yes | 127(51.8%) | 118(48.2%) | 0.97 | 0.69-1.36 | 0.846 | 0.92 | 0.69-1.23 | 0.558 |
| Metabolic syndrome | No | 200(51.2%) | 191(48.8%) | 1 | | | 1 | | |
| | Yes | 269(49.8%) | 271(50.2%) | 1.22 | 0.92-1.62 | 0.164 | 1.03 | 0.85-1.24 | 0.771 |
| Vascular events | No | 425(50.2%) | 422(49.8%) | 1 | | | 1 | | |
| | Yes | 42(51.9%) | 39(48.1%) | 0.829 | 0.49-1.41 | 0.491 | 1.055 | 0.74-1.51 | 0.770 |
| Multimorbidity | 0 diseases | 172(34.3%) | 170(33.9%) | | | | | | |
| | 1 disease | 119(23.8%) | 123(24.6%) | 1.26 | 0.87-1.81 | 0.215 | 1.09 | 0.86-1.37 | 0.486 |
| | 2 diseases | 112(22.4%) | 124(24.8%) | 1.17 | 0.80-1.72 | 0.417 | 0.99 | 0.75-1.31 | 0.942 |
| | 3-4 diseases | 98(19.6%) | 84(16.8%) | 1.04 | 0.69-1.57 | 0.858 | 1.01 | 0.77-1.33 | 0.920 |
| **Nutritional and Lab Measures** | | | | | | | | | |
| Height | continuous | | | **0.97** | **0.95-1.00** | **0.024** | **0.98** | **0.96-0.99** | **0.005** |
| Short stature | No | 265(52.9%) | 221(44.1%) | 1 | | | 1 | | |
| | Yes | 236(47.1%) | 280(55.9%) | **1.61** | **1.19-2.17** | **0.002** | **1.36** | **1.13-1.66** | **0.002** |
| BMI | continuous | | | **1.05** | **1.02-1.08** | **0.003** | **1.03** | **1.01-1.05** | **0.016** |
| BMI | Not overweight | 170(52.8%) | 152(47.2%) | 1 | | | 1 | | |
| | Overweight | 188(51.2%) | 179(48.8%) | 1.21 | 0.87-1.68 | 0.266 | 1.09 | 0.87-1.37 | 0.438 |
| | Obese | 111(45.9%) | 131(54.1%) | **1.64** | **1.10-2.45** | **0.015** | **1.35** | **1.04-1.74** | **0.022** |

*(Continued)*

**Table 3.** (Continued)

| Variables | Categories | IgE | | IgE (<vs. ≥ median) | | | IgE (continuous on log scale) | | |
|---|---|---|---|---|---|---|---|---|---|
| Double burden of malnutrition | None | 90(19.2%) | 66(14.3%) | 1 | | | 1 | | |
| | Short stature only | 80(17.1%) | 86(18.6%) | 1.25 | 0.87-1.81 | 0.229 | 1.30 | 0.77-2.19 | 0.335 |
| | Overweight/Obese only | 143(30.5%) | 116(25.1%) | 1.19 | 0.82-1.73 | 0.358 | 1.13 | 0.72-1.79 | 0.600 |
| | Both | 156(33.3%) | 194(42.0%) | 1.01 | 0.65-1.57 | 0.973 | **2.00** | **1.29-3.10** | **0.002** |
| Insulin resistance – yes | No | 179(46.5%) | 206(53.5%) | 1 | | | 1 | | |
| | Yes | 290(53.1%) | 256(46.9%) | 1.03 | 0.75-1.41 | 0.871 | 1.04 | 0.84-1.28 | 0.736 |
| WHtR elevated | No | 75(53.6%) | 65(46.4%) | 1 | | | 1 | | |
| | Yes | 394(49.8%) | 397(50.2%) | 1.51 | 1.00-2.29 | 0.053 | 1.33 | 0.99-1.78 | 0.057 |
| Elevated body fat | No | 16(53.3%) | 14(46.7%) | 1 | | | 1 | | |
| | Yes | 357(51.3%) | 339(48.7%) | **2.64** | **1.14-6.12** | **0.024** | 1.48 | 0.81-2.69 | 0.203 |
| High VAI | No | 175(48.6%) | 185(51.4%) | 1 | | | 1 | | |
| | Yes | 294(51.5%) | 277(48.5%) | 1.05 | 0.79-1.41 | 0.725 | 1.02 | 0.83-1.25 | 0.882 |
| Abdominal obesity | No | 192(50.5%) | 188(49.5%) | 1 | | | 1 | | |
| | Yes | 277(50.3%) | 274(49.7%) | 1.25 | 0.92-1.70 | 0.157 | 1.14 | 0.95-1.38 | 0.167 |
| Elevated blood cholesterol | No | 196(49.6%) | 199(50.4%) | 1 | | | 1 | | |
| | Yes | 273(50.9%) | 263(49.1%) | 1.03 | 0.76-1.38 | 0.855 | 0.99 | 0.81 | 0.931 |
| Elevated triglycerides | No | 248(49.9%) | 249(50.1%) | 1 | | | 1 | | |
| | Yes | 221(50.9%) | 213(49.1%) | 1.09 | 0.82-1.45 | 0.548 | 0.95 | 0.77-1.16 | 0.587 |
| Low HDL | No | 185(45.5%) | 222(54.5%) | 1 | | | 1 | | |
| | Yes | 284(54.2%) | 240(45.8%) | 0.99 | 0.73-1.35 | 0.960 | 0.88 | 0.72-1.09 | 0.256 |

Associations are presented for IgE analysed as a binary outcome (<140 vs. ≥ 140 IU/mL; odds ratios [ORs]) and as a continuous outcome (geometric mean ratios [GMRs]). Estimates were derived from population-weighted multilevel regression models adjusted for age, age², sex, and age–sex interaction (where significant), with clustering at household and community levels. ORs represent the adjusted odds of elevated IgE (≥140 IU/mL), and GMRs represent fold changes in IgE concentration. Abbreviations: BMI – body mass index; WHtR – waist-to-height ratio; VAI – visceral adiposity index; HDL – high-density lipoprotein. In contrast to CRP, IgE shows limited associations with cardiometabolic outcomes and likely reflects environmental and socioeconomic exposures.

to explain most of the CRP effect on blood pressure and insulin resistance. Our data indicate that systemic inflammation may be an important component of cardiometabolic risk in rural transitional communities, and that this effect may be mediated at least in part by increases in BMI. In contrast, total IgE—despite its similarly high prevalence—was not associated with cardiometabolic diseases or multimorbidity, highlighting a potential divergence between innate and type 2 inflammatory pathways in this population.

The strong and consistent associations observed between CRP and cardiometabolic diseases and multimorbidity in the Results section align with the known role of chronic low-grade systemic inflammation in promoting cardiometabolic disorders through mechanisms including endothelial dysfunction, insulin resistance, and atherogenesis [2,3,9]. Elevated CRP was associated with a range of cardiometabolic risk factors including several adiposity indicators, insulin resistance (via HOMA-IR), elevated triglycerides, and low HDL cholesterol. These associations, in accordance with the findings of previous studies [9,49,50], suggest that CRP may provide a useful proxy for cardiometabolic dysregulation in this population. Strength of CRP associations increased with number of chronic conditions present (i.e., multimorbidity). This gradient with increasing multimorbidity, clearly demonstrated in the Results, supports the role of systemic inflammation as a cumulative marker of disease burden. The consistency of associations across multiple outcomes, despite minimal adjustment, supports a robust relationship, although causality cannot be inferred. However, these patterns should be interpreted in terms of relative differences and directional associations rather than absolute numerical differences, particularly given the cross-sectional design of the study. A lack of association with vascular events may relate to possibly elevated mortality

**Table 4. Multivariable analyses of CRP and body mass index (BMI) as potential intermediate factors in pathways for biomarkers for hypertension and type-2 diabetes.**

| Exposure | Systolic blood pressure | | | | Diastolic blood pressure | | | | Glycosylated hemoglobin (HbA1c) | | | | Insulin resistance (HOMA) | | | |
|---|---|---|---|---|---|---|---|---|---|---|---|---|---|---|---|---|
| | Diff | p-value | CI-low | CI-high | Diff | p-value | CI-low | CI-high | Fold | p-value | CI-low | CI-high | Fold | p-value | CI-low | CI-high |
| Age | 0.911 | <0.001 | 0.819 | 1.002 | 0.340 | <0.001 | 0.288 | 0.392 | 1.005 | <0.001 | 1.004 | 1.006 | 1.007 | <0.001 | 1.004 | 1.010 |
| Sex | 2.237 | 0.072 | -0.199 | 4.672 | 0.043 | 0.955 | -1.459 | 1.545 | 9.589E-01 | 0.001 | 9.352E-01 | 9.822E-01 | 6.035E-01 | <0.001 | 5.353E-01 | 6.805E-01 |
| Age×Sex | -0.451 | <0.001 | -0.589 | -0.313 | -0.116 | 0.005 | -0.197 | -0.035 | 9.980E-01 | 0.001 | 9.970E-01 | 9.990E-01 | 1.000E+00 | | 1.000E+00 | 1.000E+00 |
| Age² | -3.9E-03 | 0.028 | -7.3E-03 | -4.3E-04 | -0.009 | <0.001 | -0.011 | -0.006 | 9.999E-01 | 0.002 | 9.999E-01 | 1.000E+00 | 9.998E-01 | 0.002 | 9.996E-01 | 9.999E-01 |
| CRP only | | | | | | | | | | | | | | | | |
| CRP (>3 vs. <3) | 4.296 | 0.001 | 1.754 | 6.839 | 3.688 | <0.001 | 2.024 | 5.353 | 1.057 | <0.001 | 1.033 | 1.082 | 1.489 | <0.001 | 1.327 | 1.670 |
| BMI only | | | | | | | | | | | | | | | | |
| Over-weight vs. Normal | 8.671 | <0.001 | 5.983 | 11.359 | 5.218 | <0.001 | 3.589 | 6.847 | 1.037 | 0.005 | 1.011 | 1.064 | 1.655 | <0.001 | 1.456 | 1.883 |
| Obese vs. Normal | 11.653 | <0.001 | 8.611 | 14.695 | 8.191 | <0.001 | 6.196 | 10.187 | 1.059 | <0.001 | 1.027 | 1.091 | 2.918 | <0.001 | 2.555 | 3.337 |
| BMI+CRP | | | | | | | | | | | | | | | | |
| CRP (>3 vs. <3) | 1.411 | 0.308 | -1.305 | 4.126 | 1.773 | 0.045 | 0.043 | 3.503 | 1.046 | <0.001 | 1.021 | 1.071 | 1.161 | 0.006 | 1.045 | 1.290 |
| Over-weight vs. Normal | 8.357 | <0.001 | 5.599 | 11.116 | 4.824 | <0.001 | 3.179 | 6.469 | 1.026 | 0.035 | 1.002 | 1.052 | 1.603 | <0.001 | 1.409 | 1.824 |
| Obese vs. Normal | 11.071 | <0.001 | 7.707 | 14.436 | 7.460 | <0.001 | 5.312 | 9.609 | 1.040 | 0.014 | 1.008 | 1.071 | 2.748 | <0.001 | 2.399 | 3.152 |

Associations between CRP and/or BMI and key cardiometabolic biomarkers (blood pressure, HbA1c, and insulin resistance [HOMA-IR]) were evaluated using multilevel regression models adjusted for age, age², sex, and age–sex interaction (where significant), with clustering at household and community levels. Estimates represent differences (for blood pressure outcomes) or geometric mean ratios/fold changes (for HbA1c and HOMA-IR). Models including both CRP and BMI allow assessment of attenuation of effects consistent with potential mediation pathways. BMI – body mass index. These analyses suggest that adiposity may lie upstream of CRP in pathways linking inflammation to cardiometabolic risk.

among those with both elevated CRP and advanced atherosclerotic disease who may be more likely to die at the first event [45,51] or more rapidly following a previous event [52,53].

These patterns directly mirror the Results, where CRP showed consistent associations across adiposity, metabolic risk factors, and multimorbidity. These findings should be interpreted within the context of a resource-limited primary care system, where access to diagnostic testing is restricted and cardiometabolic conditions are underdiagnosed and detected late. In such settings, simple and scalable biomarkers such as CRP may have practical value for identifying individuals at elevated risk, even in the absence of comprehensive clinical evaluation. In contrast, total IgE, while biologically informative, appears less useful as a clinical risk marker in this context.

There was not enough evidence to support associations between CRP and cardiometabolic diseases among individuals with short stature unless concurrent overweight/obesity was present, suggesting a central role of obesity in mediating the CRP effect in agreement with previous studies [49]. When adjusted for BMI, the association between CRP and insulin resistance was attenuated, indicating that BMI may lie upstream of CRP in this pathway. Conversely, there was limited evidence to suggest that BMI altered the association of CRP with HbA1c, suggesting potentially independent inflammatory contributions to glycaemic dysregulation. These findings underscore the complex interplay between adiposity and inflammation in determining cardiometabolic outcomes.

The absence of associations between total IgE and cardiometabolic diseases observed in the Results section indicates a clear divergence from CRP, despite total IgE being markedly elevated in this population, with approximately 50% of participants exceeding 140 IU/mL - above reference ranges reported in high-income settings [46]. In contrast to findings from HICs where high total IgE has been linked to allergic sensitization and cardiometabolic risk [28–31], IgE in this context likely reflects chronic exposure to helminths and other environmental pathogens [23,33,54]. Such exposures are associated with a regulated Th2 immune phenotype characterized by sustained type 2 activation alongside regulatory mechanisms that dampen pro-inflammatory pathways [34,36,55] relevant to metabolic dysfunction [35]. In this setting, therefore, total IgE is more appropriately interpreted as a marker of immune adaptation rather than dysregulation, and may not be expected to track with cardiometabolic risk in the same way as innate inflammatory markers such as CRP. From an evolutionary perspective, elevated IgE in such environments likely represents a physiologic response to persistent parasitic exposures rather than a pathologic response to innocuous allergens, and may contribute to protection against metabolic dysfunction through immunomodulatory effects [34]. These contextual differences are particularly important for interpreting total IgE, as its biological meaning may differ substantially between populations with high infectious exposure and those in high-income settings where IgE more commonly reflects allergic sensitization. Total IgE was associated with short stature, illiteracy, and obesity, suggesting links to environmental and socioeconomic exposures rather than clinical cardiometabolic outcomes.

The sex-specific differences observed in the Results, with higher CRP in women and higher IgE in men, may reflect both biological and behavioural factors. For CRP, adiposity-related differences—particularly higher central obesity and body fat percentages in women—may partly explain the observed sex disparity. In contrast, greater occupational exposure to helminth and ectoparasitic infestations among men in agricultural work may contribute to higher IgE levels. While total IgE barely changed with age, CRP levels rose until around 60–70 years, particularly in women. The difference in proportions of individuals with elevated CRP with and without multimorbidity appeared to diminish with advancing age.

The high prevalence of short stature and double burden of malnutrition observed in the Results provides important context for interpreting these findings. Adult height is recognized as the cumulative outcome of genetic and early-life environmental factors, including nutrition and infections during childhood. Short adult stature in Latin American countries including Ecuador, is largely driven by childhood stunting due to chronic undernutrition, poverty, and environmental exposures [56,57]. We used short stature as a marker for chronic malnutrition during early childhood and the combination of short stature and overweight/obesity to measure the double burden of malnutrition [57]. Our data showed that over 50% of the sample has short stature, nearly two-thirds had overweight/obesity and over a third (37.6%) suffered from the DBM. DBM,

that has been increasingly documented in LMICs including Ecuador [42,57], has been attributed to chronic undernutrition in early life followed by exposure to obesogenic environments during adulthood [57]. DBM may be particularly pronounced in rapidly urbanizing and transitional rural populations [44]. Short adult stature in this sample likely reflects childhood stunting caused by a vicious cycle of recurrent infections and chronic undernutrition, compounded by rural poverty and other structural inequalities [18,56,57]. Early childhood undernutrition and recurrent infections have been suggested to programme inflammatory and metabolic pathways in ways that predispose individuals to later-life to metabolic dysfunction and inflammation [2,58].

The high prevalence of systemic inflammation, obesity, and cardiometabolic multimorbidity observed in the results, highlights the need for context-appropriate, multidisciplinary interventions across the life course to reduce the long-term burden of cardiometabolic diseases. Such interventions may include: i) improving maternal and child nutrition; ii) reducing childhood infections through improved access to water, sanitation, and hygiene; iii) promoting healthy lifestyles in early adulthood to reduce obesity and increase physical activity; and iv) strengthening primary health-care systems for earlier diagnosis and management of cardiometabolic diseases to slow progression and minimise premature morbidity and mortality. Within this context, our findings support the usefulness of CRP as a low-cost bio-marker for identifying high-risk individuals for cardiometabolic diseases in low-resource settings. Such approaches would need to be embedded within broader efforts to strengthen primary healthcare capacity and ensure access to long-term management. These findings are likely to be most relevant to similar transitional rural populations in LMICs and underscore the need for context-specific interpretation of inflammatory biomarkers across different epidemiological settings.

Important limitations should be considered when interpreting the associations observed in the Results. Data collection was conducted in geographically restricted area in marginalized communities identified previously as having substantial unmet health needs. The generalizability of these findings is likely to be greatest for populations in transitional rural settings in LMICs undergoing rapid epidemiological and nutritional transitions, particularly in other Andean countries of Latin America. Such populations often experience a dual burden of persistent infectious exposures and increasing cardiometabolic risk, similar to the context of the present study. In contrast, the relevance of these findings to high-income settings may be more limited, given differences in environmental exposures, particularly helminth infections, as well as healthcare access and lifestyle factors. A potential source of bias is the non-participation of healthier individuals, which could have led to an overestimation of effects. Most non-participants were working-age adults (20–40 years), especially men, who were unavailable due to occupational commitments. However, the analyses were weighted using the census population to adjust for this effect. While clinical histories and risk factors were collected using questionnaires—which may be prone to recall and social desirability bias—standardized clinical and laboratory methods were employed to obtain measurements of risk factors and health outcomes wherever feasible. Blood pressure was measured using a standardized protocol; however, repeated measurements were obtained only when initial readings were elevated, rather than for all participants as recommended by WHO STEPS protocols [37]. While this approach was adopted to balance accuracy with feasibility in field conditions, it may have introduced some measurement variability and potential misclassification of hypertension status. We did not exclude participants with conditions that may influence inflammatory or metabolic biomarkers (e.g., malignancy or liver disease, likely infrequent in this population) or IgE levels (e.g., allergic, autoimmune, or parasitic conditions). In this setting, exposures that influence IgE—particularly parasitic infections—are common and likely to shape the observed IgE distribution [33]. Although analyses were adjusted for key demographic factors, residual confounding by socioeconomic status and environmental exposures cannot be excluded. These factors may influence both inflammatory markers and cardiometabolic risk in this population and should be considered when interpreting the observed associations. While CRP is a useful and robust measure for chronic systemic inflammation, other pro-inflammatory cytokines (e.g., interleukin-6, tumor necrosis factor-α) could offer a better understanding of the nature and mechanisms underlying systemic inflammation in this population. The cross-sectional nature of the study does not allow us to clearly understand

causal pathways for the cardiometabolic diseases measured – longitudinal studies are required to disentangle temporal relationships between systemic inflammation, nutritional status, cardiometabolic risk and potential causal relationships.

## Conclusion

Our findings demonstrate a high prevalence of systemic inflammation and cardiometabolic risk factors and associated diseases among adults living in transitional rural communities in Ecuador. Elevated CRP was strong associated with cardiometabolic risk and multimorbidity. While elevated total IgE was associated with poor nutritional status, links to disease outcomes were less clear, and in the study setting likely reflected a regulated Th2 immune response to parasites. Body mass index explained better blood pressure and insulin resistance in this population, while levels of CRP appeared to be largely determined by the presence of a high BMI. Our findings point to a high prevalence of systemic inflammation, cardiometabolic risk factors, and diseases among adults living in rural agricultural communities in Ecuador. Future studies will need to explore longitudinally the natural history of systemic inflammation, its causes, and its temporal link to obesity to identify potential interventions that might be implemented to ameliorate an epidemic of premature morbidity and mortality that has emerged in low resource non-industrialized populations undergoing rapid epidemiologic and nutritional transitions.

## Supporting information

**S1 Fig. The distribution of the history of vascular events overall (A) and by sex (B) in the sample population.**
(TIFF)

**S2 Fig. Observed and estimated age-specific CRP (A) and IgE (B) levels at log scale, stratified by sex.**
(TIFF)

**S1 Data. Raw data used for analyses.**
(XLS)

## Acknowledgments

The authors thank participant communities and their representatives, particularly Sr Efren Velez, for their co-operation and support, and the support also of technicians and health professionals from Universidad Internacional del Ecuador and the CAMERA study.

## Author contributions

**Conceptualization:** Natalia Romero-Sandoval, Philip J. Cooper.

**Data curation:** Monsermin Gualan.

**Formal analysis:** Irina Chis Ster.

**Funding acquisition:** Natalia Romero-Sandoval, Philip J. Cooper.

**Investigation:** Monsermin Gualan, Luz-Marina Llangari-Arizo, Andrea Lopez.

**Methodology:** Philip J. Cooper.

**Project administration:** Natalia Romero-Sandoval, Philip J. Cooper.

**Resources:** Philip J. Cooper.

**Supervision:** Luz-Marina Llangari-Arizo, Natalia Romero-Sandoval, Philip J. Cooper.

**Visualization:** Irina Chis Ster.

**Writing – original draft:** Irina Chis Ster, Philip J. Cooper.

**Writing – review & editing:** Monsermin Gualan, Luz-Marina Llangari-Arizo, Andrea Lopez, Natalia Romero-Sandoval.

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
