## [Decision Letter · Decision Letter 0]

11 Mar 2026

PONE-D-25-57953Associations between systemic inflammation, nutritional status, and cardiometabolic diseases and risk factors among adults living in transitional rural communities in Ecuador.PLOS One

Dear Dr. Cooper,

Thank you for submitting your manuscript to PLOS ONE. After careful consideration, we feel that it has merit but does not fully meet PLOS ONE’s publication criteria as it currently stands. Therefore, we invite you to submit a revised version of the manuscript that addresses the points raised during the review process.

We look forward to receiving your revised manuscript.

Kind regards,

Swapnil Parve, M.D., Ph.D.

Academic Editor

PLOS One

Journal Requirements:

2. We notice that your supplementary figures are uploaded with the file type 'Figure'. Please amend the file type to 'Supporting Information'. Please ensure that each Supporting Information file has a legend listed in the manuscript after the references list.

Additional Editor Comments:

For your convenience I have attached the reviewer comments to this email.

Reviewer 1 comments:

Comments:

The captions for figures and tables should be more informative to enhance reader understanding. Clearer descriptions of what each figure and table represents will aid in interpreting the data presented.

The authors should provide details regarding the clinical settings in which their findings were utilized. This information is crucial for interpreting risk factors and drawing meaningful conclusions from the results.

I encourage the authors to consider performing external validation of their findings using additional databases. This step would strengthen the robustness of their conclusions and enhance the generalizability of the results.

Please include specific information about the study sample size in the results section. Knowing the sample size is essential for assessing the reliability and validity of the study's findings.

The keywords should be formatted according to MeSH (Medical Subject Headings) standards. Properly selected keywords will improve the discoverability of the article in relevant searches.

It is important to clarify the age range covered by the various data used in this study. This information will help contextualize the findings within specific demographic groups.

I recommend restructuring the results section in a more comparative manner. This approach will facilitate a clearer understanding of differences between groups and highlight key findings more effectively.

All abbreviations should be defined upon first use in both the abstract and main text. This practice will improve clarity for readers who may not be familiar with specific terms.

Research involving human participants must comply with the Helsinki Declaration. Please include a statement clarifying whether this study adhered to the Declaration of Helsinki in the ‘Ethics approval and consent to participate’ section.

The methods section should provide a comprehensive overview of the data collection process, including specific steps taken to ensure accuracy and reliability.

The authors should clarify how effectively they controlled for confounding factors in their analysis.

The authors should elaborate on how their findings can be generalized to other populations or regions with different contexts.

To enhance clarity, it is recommended that the authors include a methodological flowchart or workflow figure.

The results section should be more effectively integrated into the discussion.

It is important to emphasize that interpretations of long-term trends should focus on relative directional changes rather than absolute numerical differences. The authors should clarify this point in their discussion to avoid misinterpretation of data over time.

The authors should clarify whether any adjustments were made for underreporting or data gaps in certain locations.

Reviewer 2 comments:

The study titled “Associations between systemic inflammation, nutritional status, and cardiometabolic diseases and risk factors among adults living in transitional rural communities in Ecuador” by Ster et al. addresses an important topic. However, to strengthen the manuscript and improve clarity, I recommend the following edits:

1. Pages 12-14: The introduction would benefit from a clearer emphasis on the importance of studying IgE in this specific population. While the manuscript discusses the potential role of IgE in cardiometabolic disease, the rationale for examining IgE in this setting is not sufficiently developed.

2. Page 17, Line 165: The authors define elevated CRP as ≥3 mg/dL. This threshold appears inconsistent with commonly used definitions in the literature, where CRP cut-offs for cardiometabolic risk are typically reported in mg/L rather than mg/dL. The authors should recheck the CRP units and cut-off, as this may represent a unit error.

3. Page 17, Line 166: The definition of elevated IgE as the median value (140 IU/mL) is unclear and requires clarification. The authors should describe this more clearly and justify the rationale for using the median as a cut-off.

4. Page 16, Lines 126-128: The manuscript states that a second blood pressure reading was taken only if SBP ≥130 mmHg or DBP ≥85 mmHg. This approach should be clarified, as standard protocols typically recommend multiple readings for all participants, not just the ones with higher BP.

5. Page 15, Lines 115-116: The start date of participant recruitment is incomplete, as the year is not specified. The authors should provide the full dates to ensure clarity and accuracy in reporting the study timeline.

6. Pages 15-19: The Methods section does not specify whether participants with chronic conditions that could significantly alter metabolic biomarker levels (e.g., cancer or advanced liver disease) were excluded. The authors should clarify this to ensure appropriate interpretation of the biomarker data.

7. Pages 15-19: The Discussion emphasizes that elevated IgE in this population primarily reflects physiologic responses to environmental pathogens rather than pathologic allergic responses. However, the Methods section does not specify whether participants with conditions that could confound IgE levels, such as asthma, autoimmune diseases, allergic dermatitis, other allergic conditions, or parasitic infections, were excluded. The authors should clarify the inclusion/exclusion criteria regarding these conditions to ensure proper interpretation of IgE findings.

Reviewer's Responses to Questions

**Comments to the Author**

1. Is the manuscript technically sound, and do the data support the conclusions?

Reviewer #1: Yes

Reviewer #2: Yes

2. Has the statistical analysis been performed appropriately and rigorously? 

Reviewer #1: Yes

Reviewer #2: Yes

3. Have the authors made all data underlying the findings in their manuscript fully available?

Reviewer #1: Yes

Reviewer #2: Yes

4. Is the manuscript presented in an intelligible fashion and written in standard English?

Reviewer #1: Yes

Reviewer #2: Yes

5. Review Comments to the Author

Reviewer #1: The manuscript investigates the associations between systemic inflammation, as indicated by high-sensitivity C-reactive protein (hs-CRP) and total immunoglobulin E (IgE), and cardiometabolic diseases and risk factors among adults in transitional rural communities in coastal Ecuador. The authors conducted a cross-sectional analysis involving 931 adults from ten agricultural communities, utilizing standardized questionnaires, anthropometric measurements, and biochemical assessments to collect data on sociodemographics, body composition, inflammatory markers, and cardiometabolic outcomes. The study finds a significant prevalence of elevated CRP levels and their associations with hypertension, type 2 diabetes, metabolic syndrome, and multimorbidity, while total IgE does not appear to correlate with cardiometabolic diseases.

Comments:

The captions for figures and tables should be more informative to enhance reader understanding. Clearer descriptions of what each figure and table represents will aid in interpreting the data presented.

The authors should provide details regarding the clinical settings in which their findings were utilized. This information is crucial for interpreting risk factors and drawing meaningful conclusions from the results.

I encourage the authors to consider performing external validation of their findings using additional databases. This step would strengthen the robustness of their conclusions and enhance the generalizability of the results.

Please include specific information about the study sample size in the results section. Knowing the sample size is essential for assessing the reliability and validity of the study's findings.

The keywords should be formatted according to MeSH (Medical Subject Headings) standards. Properly selected keywords will improve the discoverability of the article in relevant searches.

It is important to clarify the age range covered by the various data used in this study. This information will help contextualize the findings within specific demographic groups.

I recommend restructuring the results section in a more comparative manner. This approach will facilitate a clearer understanding of differences between groups and highlight key findings more effectively.

All abbreviations should be defined upon first use in both the abstract and main text. This practice will improve clarity for readers who may not be familiar with specific terms.

Research involving human participants must comply with the Helsinki Declaration. Please include a statement clarifying whether this study adhered to the Declaration of Helsinki in the ‘Ethics approval and consent to participate’ section.

The methods section should provide a comprehensive overview of the data collection process, including specific steps taken to ensure accuracy and reliability.

The authors should clarify how effectively they controlled for confounding factors in their analysis.

The authors should elaborate on how their findings can be generalized to other populations or regions with different contexts.

To enhance clarity, it is recommended that the authors include a methodological flowchart or workflow figure.

The results section should be more effectively integrated into the discussion.

It is important to emphasize that interpretations of long-term trends should focus on relative directional changes rather than absolute numerical differences. The authors should clarify this point in their discussion to avoid misinterpretation of data over time.

The authors should clarify whether any adjustments were made for underreporting or data gaps in certain locations.

Sincerely,

Reviewer #2: The study titled “Associations between systemic inflammation, nutritional status, and cardiometabolic diseases and risk factors among adults living in transitional rural communities in Ecuador” by Ster et al. addresses an important topic. However, to strengthen the manuscript and improve clarity, I recommend the following edits:

1. Pages 12-14: The introduction would benefit from a clearer emphasis on the importance of studying IgE in this specific population. While the manuscript discusses the potential role of IgE in cardiometabolic disease, the rationale for examining IgE in this setting is not sufficiently developed.

2. Page 17, Line 165: The authors define elevated CRP as ≥3 mg/dL. This threshold appears inconsistent with commonly used definitions in the literature, where CRP cut-offs for cardiometabolic risk are typically reported in mg/L rather than mg/dL. The authors should recheck the CRP units and cut-off, as this may represent a unit error.

3. Page 17, Line 166: The definition of elevated IgE as the median value (140 IU/mL) is unclear and requires clarification. The authors should describe this more clearly and justify the rationale for using the median as a cut-off.

4. Page 16, Lines 126-128: The manuscript states that a second blood pressure reading was taken only if SBP ≥130 mmHg or DBP ≥85 mmHg. This approach should be clarified, as standard protocols typically recommend multiple readings for all participants, not just the ones with higher BP.

5. Page 15, Lines 115-116: The start date of participant recruitment is incomplete, as the year is not specified. The authors should provide the full dates to ensure clarity and accuracy in reporting the study timeline.

6. Pages 15-19: The Methods section does not specify whether participants with chronic conditions that could significantly alter metabolic biomarker levels (e.g., cancer or advanced liver disease) were excluded. The authors should clarify this to ensure appropriate interpretation of the biomarker data.

7. Pages 15-19: The Discussion emphasizes that elevated IgE in this population primarily reflects physiologic responses to environmental pathogens rather than pathologic allergic responses. However, the Methods section does not specify whether participants with conditions that could confound IgE levels, such as asthma, autoimmune diseases, allergic dermatitis, other allergic conditions, or parasitic infections, were excluded. The authors should clarify the inclusion/exclusion criteria regarding these conditions to ensure proper interpretation of IgE findings.

6. PLOS authors have the option to publish the peer review history of their article (what does this mean?). If published, this will include your full peer review and any attached files.

Reviewer #1: No

Reviewer #2: No

---

## [Author Response · Author response to Decision Letter 1]

21 Apr 2026

Dear Professor Parve,

We would like to thank the Academic Editor and both reviewers for their careful reading of our manuscript and for their thoughtful, constructive, and insightful comments. We greatly appreciate the time and expertise invested in reviewing our work. We have addressed all comments in detail below and have revised the manuscript accordingly. We believe these revisions have significantly improved the clarity, rigor, and overall quality of the manuscript.

Reviewer 1 comments:

Comments:

Comment 1: The captions for figures and tables should be more informative to enhance reader understanding. Clearer descriptions of what each figure and table represents will aid in interpreting the data presented.

Response 1: We thank the reviewer for this helpful suggestion. We have revised all figure and table captions to improve clarity and interpretability. Specifically, captions now clearly describe what is being shown and the purpose of each figure/table, specify key definitions (e.g., biomarker cut-offs), indicate the statistical methods used, clarify denominators and note missing data where relevant, and provide guidance on how to interpret estimates (e.g., odds ratios vs. geometric mean ratios) These revisions ensure that figures and tables are self-contained and can be interpreted without reference to the main text.

Comment 2: The authors should provide details regarding the clinical settings in which their findings were utilized. This information is crucial for interpreting risk factors and drawing meaningful conclusions from the results.

Response 2: We thank the reviewer for this important comment. We have clarified the clinical and health system context of the study in both the Methods and Discussion sections. Specifically, we now describe the rural primary care setting in which the study was conducted, including limited access to diagnostic services and referral care. We emphasize that this was a population-based epidemiological investigation rather than a clinical study, and that findings were not applied within a clinical decision-making framework. In addition, we have strengthened the Discussion to explicitly interpret our findings in relation to this clinical context. We highlight the potential utility of simple biomarkers such as CRP for risk stratification in low-resource primary care settings, while noting the more limited clinical relevance of total IgE in this context. These revisions clarify how the findings relate to real-world healthcare delivery and their potential implications for screening and prevention strategies in underserved populations.

Comment 3: I encourage the authors to consider performing external validation of their findings using additional databases. This step would strengthen the robustness of their conclusions and enhance the generalizability of the results.

Response 3: We agree that external validation would strengthen the generalizability of the findings. However, the current study was designed as an exploratory, hypothesis-generating analysis in a unique transitional rural population for which comparable datasets are not readily available, particularly datasets including hs-CRP and IgE measurements. We have now acknowledged this limitation explicitly in the Discussion and highlighted external validation in other LMIC populations as an important priority for future research.

Comment 4: Please include specific information about the study sample size in the results section. Knowing the sample size is essential for assessing the reliability and validity of the study's findings.

Response 4: We thank the reviewer for this comment. We have now explicitly stated the study sample size (n = 931) at the beginning of the Results section.

Comment 5: The keywords should be formatted according to MeSH (Medical Subject Headings) standards. Properly selected keywords will improve the discoverability of the article in relevant searches.

Response 5: We thank the reviewer for this suggestion. We have revised the keywords to align with Medical Subject Headings (MeSH) terminology to improve indexing and discoverability. Specifically, we have used the following MeSH terms: “C-Reactive Protein,” “Immunoglobulin E,” “Inflammation,” “Metabolic Syndrome,” “Cardiovascular Diseases,” and “Rural Population.”

Comment 6: It is important to clarify the age range covered by the various data used in this study. This information will help contextualize the findings within specific demographic groups.

Response 6: We have clarified that the study population comprised adults aged ≥18 years (with no upper age limit) and have ensured this is consistently stated in the Methods, Results, and relevant table and figure legends. This improves the contextualization of findings across the adult age range included in the analysis.

Comment 7: I recommend restructuring the results section in a more comparative manner. This approach will facilitate a clearer understanding of differences between groups and highlight key findings more effectively.

Response 7: We thank the reviewer for this helpful suggestion. We have revised the Results section to improve comparative clarity and highlight key differences between groups as follows: 1) introduced clearer comparative framing in the text (e.g., by sex, biomarker type, and disease status); ii) strengthened direct contrasts between CRP and total IgE findings; iii) added interpretive sentences to highlight key differences in associations with cardiometabolic outcomes and risk factors; and iv) included a brief summary paragraph to synthesize the main comparative findings These revisions improve readability and make the contrasting patterns between inflammatory markers and population subgroups more explicit.

Comment 8: All abbreviations should be defined upon first use in both the abstract and main text. This practice will improve clarity for readers who may not be familiar with specific terms.

Response 8: All abbreviations are now defined at first use in both the Abstract and main text.

Comment 9: Research involving human participants must comply with the Helsinki Declaration. Please include a statement clarifying whether this study adhered to the Declaration of Helsinki in the ‘Ethics approval and consent to participate’ section.

Response 9: We thank the reviewer for this important comment. We have now included a statement in the Ethics statement section confirming that all procedures involving human participants were conducted in accordance with the 1964 Helsinki Declaration and its later amendments.

Comment 10: The methods section should provide a comprehensive overview of the data collection process, including specific steps taken to ensure accuracy and reliability.

Response 10: We have strengthened the Methods section by explicitly detailing procedures implemented to ensure data quality, including staff training, standardized measurement protocols, duplicate measurements, laboratory quality control, and data validation checks.

Comment 11: The authors should clarify how effectively they controlled for confounding factors in their analysis.

Response 11: We have clarified our approach to confounding in both the Methods and Discussion sections. Analyses were adjusted for age, sex, and their interaction, which are key determinants of both inflammatory markers and cardiometabolic outcomes. We adopted a parsimonious adjustment strategy to avoid overadjustment, particularly for variables such as body mass index (BMI), which may lie on the causal pathway between systemic inflammation and cardiometabolic disease. We have clarified our approach to confounding in the Methods and Discussion. Analyses were minimally adjusted for age, sex, and age–sex interaction to avoid overadjustment, particularly where variables such as BMI may lie on the causal pathway. We now explicitly discuss the potential for residual confounding (e.g., socioeconomic and environmental factors) and interpret findings cautiously as exploratory associations.

Comment 12: The authors should elaborate on how their findings can be generalized to other populations or regions with different contexts.

Response 12: We thank the reviewer for this important comment. We have expanded the Discussion to clarify the generalizability of our findings. Specifically, we now emphasize that the findings are most applicable to populations in transitional rural settings in low- and middle-income countries undergoing rapid epidemiological and nutritional transitions. These settings share key characteristics with our study population, including a high burden of infectious exposures alongside increasing prevalence of obesity and cardiometabolic diseases. We also highlight that differences in environmental exposures (e.g., helminth infections), healthcare access, and lifestyle factors may influence the extent to which these findings can be generalized to other populations, particularly those in high-income settings. These additions provide clearer context for interpreting the broader relevance of our results.

Comment 13: To enhance clarity, it is recommended that the authors include a methodological flowchart or workflow figure.

Response 13: We have added a methodological flow diagram (new Figure 1) as requested.

Comment 14: The results section should be more effectively integrated into the discussion.

Response 14: We thank the reviewer for this helpful suggestion. We have revised the Discussion to more closely integrate the Results and improve the continuity between sections. Specifically, we have: i) added explicit references to key findings from the Results (e.g., CRP–multimorbidity associations, lack of IgE associations, and sex-specific patterns); ii) strengthened the interpretation of these findings within relevant biological and clinical frameworks; and iii) ensured that each Discussion paragraph is clearly anchored to corresponding Results. These revisions improve the coherence of the manuscript and make the interpretation of findings more directly linked to the results presented.

Comment 15: It is important to emphasize that interpretations of long-term trends should focus on relative directional changes rather than absolute numerical differences. The authors should clarify this point in their discussion to avoid misinterpretation of data over time.

Response 15: We thank the reviewer for this helpful comment. Although our study is cross-sectional and does not analyse temporal trends, we agree that care is needed in interpreting patterns across age groups and disease strata. We have clarified in the Discussion that observed patterns should be interpreted in terms of relative differences and directional associations rather than absolute numerical differences, to avoid overinterpretation.

Comment 16: The authors should clarify whether any adjustments were made for underreporting or data gaps in certain locations.

Response 16: We clarify that data were collected through direct field measurements and standardized questionnaires rather than secondary datasets, minimizing issues of underreporting. However, we now acknowledge potential non-participation bias and its possible effects in the Discussion. Post-stratification weighting techniques using population distributions by age and sex from Census data was used to address sampling imbalances - the closest methodology to mimicking population survey data.

Reviewer 2 comments:

The study titled “Associations between systemic inflammation, nutritional status, and cardiometabolic diseases and risk factors among adults living in transitional rural communities in Ecuador” by Ster et al. addresses an important topic. However, to strengthen the manuscript and improve clarity, I recommend the following edits:

Comment 17: 1. Pages 12-14: The introduction would benefit from a clearer emphasis on the importance of studying IgE in this specific population. While the manuscript discusses the potential role of IgE in cardiometabolic disease, the rationale for examining IgE in this setting is not sufficiently developed.

Response 17: We thank the reviewer for this helpful comment. We have revised the Introduction to more clearly articulate the rationale for studying total IgE in this population. Specifically, we now emphasize that in tropical rural settings such as the study communities, total IgE levels are typically elevated due to chronic exposure to helminths and other environmental pathogens, reflecting a regulated type 2 immune response rather than allergic sensitization. This distinct immunological context differs fundamentally from that in high-income settings, where IgE is more commonly linked to atopy. We further clarify that this provides a unique opportunity to examine whether IgE, as a marker of type 2 inflammation, is associated with cardiometabolic risk in a setting where its biological meaning differs. These revisions strengthen the rationale for including IgE and highlight the importance of studying this marker in populations undergoing epidemiological and nutritional transition.

Comment 18: 2. Page 17, Line 165: The authors define elevated CRP as ≥3 mg/dL. This threshold appears inconsistent with commonly used definitions in the literature, where CRP cut-offs for cardiometabolic risk are typically reported in mg/L rather than mg/dL. The authors should recheck the CRP units and cut-off, as this may represent a unit error.

Response 18: We thank the reviewer for identifying this important issue.

The correct unit is mg/L, not mg/dL. This has been corrected throughout the manuscript, and the cut-off (≥3 mg/L) remains appropriate and unchanged.

Comment 19: 3. Page 17, Line 166: The definition of elevated IgE as the median value (140 IU/mL) is unclear and requires clarification. The authors should describe this more clearly and justify the rationale for using the median as a cut-off.

Response 19: We thank the reviewer for this comment. We have clarified that the cut-off of 140 IU/mL corresponds to the median value in this study population and was used to define elevated IgE for analytical purposes, given the absence of standardized cut-offs in comparable populations with high baseline IgE levels. We now explicitly justify this approach as appropriate for exploratory analyses in this setting and have ensured transparency by presenting IgE both as a binary variable (using the median cut-off) and as a continuous measure.

Comment 20: 4. Page 16, Lines 126-128: The manuscript states that a second blood pressure reading was taken only if SBP ≥130 mmHg or DBP ≥85 mmHg. This approach should be clarified, as standard protocols typically recommend multiple readings for all participants, not just the ones with higher BP.

Response 20: We thank the reviewer for this important point. We have clarified the blood pressure measurement protocol in the Methods section. While WHO STEPS protocols recommend multiple measurements for all participants, in this field-based study a second reading was obtained when elevated values were detected to balance accuracy with feasibility constraints. We have also acknowledged this as a limitation in the Discussion.

Comment 21: 5. Page 15, Lines 115-116: The start date of participant recruitment is incomplete, as the year is not specified. The authors should provide the full dates to ensure clarity and accuracy in reporting the study timeline.

Response 21: We have corrected and completed the recruitment dates (31st July 2021 to 12th September 2021) in the Methods section.

Comment 22: 6. Pages 15-19: The Methods section does not specify whether participants with chronic conditions that could significantly alter metabolic biomarker levels (e.g., cancer or advanced liver disease) were excluded. The authors should clarify this to ensure appropriate interpretation of the biomarker data.

Response 22: We thank the reviewer for this important comment. The study was designed as a population-based investigation, and participants were not excluded on the basis of pre-existing chronic conditions such as cancer or liver disease. We have clarified this in the Methods section. We agree that such conditions could potentially influence biomarker levels; however, their prevalence in this setting is likely to be low, and our obje

---

## [Editor Report · Decision Letter 1]

12 May 2026

Associations between systemic inflammation, nutritional status, and cardiometabolic diseases and risk factors among adults living in transitional rural communities in Ecuador.

PONE-D-25-57953R1

Dear Dr. Cooper,

We’re pleased to inform you that your manuscript has been judged scientifically suitable for publication and will be formally accepted for publication once it meets all outstanding technical requirements.

Kind regards,

Swapnil Parve, M.D., Ph.D.

Academic Editor

PLOS One

---

## [Editor Report · Acceptance letter]

PONE-D-25-57953R1

PLOS One

Dear Dr. Cooper,

I'm pleased to inform you that your manuscript has been deemed suitable for publication in PLOS One. Congratulations! Your manuscript is now being handed over to our production team.

Kind regards,

on behalf of

Dr. Swapnil Parve

Academic Editor

PLOS One